# Effect of airway clearance therapies on mucociliary clearance in adults with cystic fibrosis: A randomized controlled trial

**Aaron Trimble**[1,2]*, **Kirby Zeman**[2], **Jihong Wu**[2], **Agathe Ceppe**[2], **William Bennett**[2], **Scott Donaldson**[2]

**1** Department of Medicine, Oregon Health and Science University (OHSU), Portland, Oregon, United States of America, **2** Department of Medicine, University of North Carolina (UNC), Chapel Hill, North Carolina, United States of America

* trimblea@ohsu.edu

**Data Availability Statement:** All relevant data are within the paper and its Supporting Information files.

## Abstract

### Background

Cystic fibrosis (CF) is an inherited disorder causing impaired mucociliary clearance within the respiratory tract, and is associated with bronchiectasis, chronic respiratory infections, and early death. Airway clearance therapies have long been a cornerstone of management of individuals with CF, although evidence supporting their use is lacking. We designed a randomized controlled trial to quantitatively compare the effects of different forms of airway clearance on mucociliary clearance.

### Methods

Three different physiotherapy methods to augment cough-clearance were studied in addition to cough-clearance alone: high-frequency chest-wall oscillating vest, oscillatory positive expiratory pressure, and whole-body vibration. We used gamma scintigraphy after inhalation of radiolabeled particles to quantify mucus clearance before, during, and after physiotherapy. As secondary endpoints, we measured concentrations of small molecules in exhaled breath that may impact mucus clearance.

### Results

Ten subjects were enrolled and completed study procedures. No differences were identified between any method of airway clearance, including cough clearance alone. We did identify changes in certain small molecule concentrations in exhaled breath following airway clearance.

### Conclusions

Due to the limitations of this study, we do not believe the negative results suggest a change in clinical practice with regard to airway clearance. Findings pertaining to small molecules in exhaled breath may serve as future opportunities for study.

**Funding:** This work was supported by a grant to AT and SD from the Cystic Fibrosis Foundation (http://www.cff.org REACT16A0). The funders of this study were not involved in the study design, execution of the procedures, data analysis, preparation of the manuscript, or decision to publish. Investigators received no financial support from commercial entities for the completion of this study. A PowerPlate® device was provided by the manufacturer, Performance Health Systems LLC, Northbrook, IL, USA.

**Competing interests:** The authors received a PowerPlate® device from the manufacturer (Performance Health Systems LLC, Northbrook, IL, USA). This product was provided for the purposes of conducting this study without stipulation or conditions, and returned following completion of the study. No other goods, services, or any other form of compensation were provided to the authors by this entity. This does not alter our adherence to PLOS ONE policies on sharing data and materials.

## Introduction

Cystic fibrosis (CF) is a multisystem genetic disease principally characterized by lung disease with thick respiratory secretions and poor mucociliary clearance (MCC), chronic airway infections, and recurrent exacerbations [1–4]. While there are several pharmacologic strategies aimed at improving CF lung disease, airway clearance therapies (ACTs) continue to be a cornerstone of routine care [5, 6]. Many methods of ACT exist and include the use of mechanical devices that deliver pressure waves to the thoracic cavity, respiratory exercises that utilize variations in air flow and tidal breathing outside of normal respiratory patterns, and devices that provide positive expiratory pressure with or without pressure oscillations [7–10]. Exercise is also commonly used as an adjunctive ACT. However, there is little evidence to suggest superiority of one method of ACT over another, and while current guidelines in Europe, the UK, and the US all advocate for ACT, the current consensus is that the specific ACT method should be tailored to the individual patient with consideration for what is practical, acceptable to the patient, economically feasible, and subjectively effective [5]. A recently updated meta-analysis from the Cochrane Library reviewed 39 studies and did not identify clear evidence to support the favorability of any ACT technique over another based on several endpoints, including pulmonary function testing, sputum weights (wet or dry), exacerbation frequency, or participant satisfaction [11].

The mucociliary and cough clearance (MCC/CC) defect in CF has been extensively studied using gamma scintigraphy methods [12–14]. This outcome measure has been used to characterize the effect of several potential CF therapies on MCC/CC, and has shown good correlation between MCC/CC improvements and clinical benefits [15–17]. Interestingly, an early study used the MCC/CC technique to study mechanical ACTs in CF and failed to demonstrate differences in measured MCC/CC between postural drainage, positive expiratory pressure, and exercise in individuals with CF [18]. Given the improvements in the technique used for MCC/CC measurements over the last 25 years, as well as the ongoing question whether some ACTs may be superior to others, we designed a multiple crossover study of the two most commonly-used techniques in the United States as well as whole body vibration and huff-coughing alone as a control. We included whole body vibration via a vibrating platform both to evaluate the potential of this novel device as an ACT and as a control for the effects of strong mechanical stimuli not specifically targeted to the lung or developed as an ACT modality. Importantly, each of these ACTs were performed in conjunction with huff coughing, which was studied alone as a baseline comparator.

Finally, we explored the effect of ACTs on small-molecule biomarkers, including fraction of exhaled nitric oxide (FeNO) and the concentrations of other potential MCC regulators in exhaled breath condensate [19–22]. Decreased FeNO has been associated with increased disease activity in CF, and increased values have been associated with treatment [19, 20]. Metabolomic studies of airway surface biology have implicated a number of small molecules present in exhaled breath condensate as associated with regulation of airway surface liquid, most notably adenosine and other purine metabolites. Other small molecules have been associated with airway inflammation, mucus clearance activity, energy and function regulation, although their roles are poorly understood. These include some nucleosides and amino acids, small peptides, the polyamine spermine, lactate, and the mucin component sialic acid.

Our primary hypothesis was that the high-frequency chest wall oscillatory vest (HFCWO-vest) and oscillatory positive expiratory pressure (OPEP) devices would accelerate MCC/CC when compared to huff coughing alone. Our secondary hypotheses were that adenosine concentrations would increase in exhaled breath condensate as the result of mechanically-stimulated ATP release and subsequent metabolism, and that FeNO would increase in parallel with improvements in MCC.

## Methods

### Study design & research setting

We conducted a pilot open-label multiple crossover study to measure the effect of airway clearance therapy (ACT) on MCC in adults with CF. Subjects were recruited from and study procedures performed at the adult CF center at the University of North Carolina. The study was reviewed by the institutional review board at UNC and registered with ClinicalTrials.gov (NCT03078127), although there was a slight delay in submission to this registry which occured after subject enrollment began. This occurred due to institutional delays in assisting with registry submission. The authors confirm that all ongoing and related trials for this intervention are registered.

### Subject eligibility

Subjects were eligible for enrollment if they were $\geq$18 years of age and had confirmed CF and a forced expiratory volume in one second ($FEV_1$) > 30% of predicted ($ppFEV_1$). Subjects were excluded if they were pregnant, had a recent exacerbation of lung disease requiring antibiotics within four weeks prior enrollment, or were unable to perform any of the ACT modalities.

### Interventions

Subjects completed a screening visit during which eligibility criteria were confirmed and study techniques were taught and reviewed, and subjects provided written informed consent for study participation. Following this, study interventions occurred over four separate study visits, with a separate ACT modality studied at each visit. Study visits were between 3 and 21 days apart to allow for washout of ACT effects while maintaining study timeliness to prevent effects from natural disease activity. Use of hypertonic saline, dornase alfa, long-acting bronchodilators, and routine airway clearance were withheld for 12 hours prior to procedures on each study visit day. Each of three ACT modalities was studied in conjunction with prescribed huff coughing maneuvers, and a huff coughing-only study visit served as a baseline comparator. The studied ACT modalities were oscillatory positive expiratory pressure (OPEP; Aerobika®, Trudell Medical International, London, ON, Canada), high-frequency chest wall oscillation (HFCWO; The Vest®, Hill-Rom Advanced Respiratory Inc., St. Paul, MN, United States), and whole body vibration (WBV; Power Plate®, Performance Health Systems LLC, Northbrook, IL, USA). A Power Plate®, device was provided by Performance Health Systems for the execution of this study. The baseline visit was completed first, followed by three visits utilizing each of OPEP, WBV, or HFCWO, performed in a random order.

At each of the study visits, subjects would inhale radiotracer used for the primary outcome measure, and then at specified time points perform ACT procedures followed by a huff cough maneuver. Each study visit consisted of two 34-minute ACT sessions carefully timed with image acquisition for the primary outcome measure. Further details on these procedures is provided in S5 File.

### Outcome measures

The primary endpoint was MCC as measured by the percent clearance of an inhaled radiolabeled marker from the whole right lung region over the 274 minutes that followed its administration. MCC was measured over 4.5 hours before, during, and after ACT sessions using a standardized methodology that involves inhalation of aerosolized $Tc^{99m}$ labelled sulfur colloid particles and collection of serial images over time with gamma scintigraphy, as previously described [23]. Further details on this method are described in the online supplement.

The primary outcome for this study was the average clearance (expressed as % and denoted Ave274Clr) of Tc$^{99m}$ activity measured within the whole right lung region of interest (ROI) from time 0 through 274 minutes in two 90-minute sessions during which ACT was performed. This was determined by averaging the percent clearance of Tc$^{99m}$ activity within the ROI from key images every 10 minutes, and is representative of the area under the curve of isotope clearance over time. Secondary MCC outcomes included isotope clearance from the whole lung region through 90 minutes (i.e. one ACT session), clearance from the peripheral and central ROI through 90 and 274 minutes, and isotope deposition indices (skew and central to peripheral ratio, C/P).

As part of this pilot study, we measured FeNO and collected exhaled breath condensates for measurement of small molecules before and after ACT at each study visit. We measured FeNO using a commercially-available device (NIOX MINO®, Circassia, Oxford, UK) according to the manufacturer's instructions, and three readings were obtained and averaged. FeNO measurements were then repeated immediately after the first ACT session. Similarly, exhaled breath condensate (EBC) was collected using the R-Tube® system to non-invasively measure small-molecule concentrations in the deep lung. Subjects breathed through the R-tube device for 7 minutes to allow condensate collection, which was immediately frozen for subsequent analysis. Mass-spectrometry, using previously-described methods, was used to measure concentrations of selected small molecules implicated in CF disease activity and/or regulation of MCC as well as urea, which was used for normalization to account for variability in sample dilution [21, 22, 24, 25]. Concentrations were reported as a unitless index determined by the ratio of measured concentration of the small molecule to the measured concentration urea in the same sample. The studied compounds are listed in the results section. Additionally, all sputum expectorated during the study sessions was collected for total weight and % solid measurement.

## Sample size determination

No comparable preliminary data on the effect of ACT on MCC was available to perform sample size calculation, as the only previous study used a different methodology for measuring MCC. However, our lab has collected pertinent data on the effects of inhaled 7% hypertonic saline on MCC using the same methods on 11 subjects with CF using a similar crossover study design, and calculated average mucus clearance over 180 minutes and 360 minutes respectively. From these data, we interpolated an estimated average clearance over 274 minutes (Ave274Clr), the primary endpoint for our study. At baseline, Ave274Clr was 32.4% (SD 19.0), and following hypertonic saline was 60.2% (SD 19.1%), with a mean difference of 27.7% (SD 15.1). We predicted that the observed mean change in Ave274Clr after ACT would be somewhat smaller compared to HS. We estimated a more conservative predicted effect size of 20.0%, with the same estimate of variability (SD 15.1). Using these data, a within-subject dependent effect size index, $d_z$ of 1.33 was calculated for Ave274Clr. Using G*Power (v3.1) and a paired, 2-tailed t-test to compare two means, we estimated that 7 subjects would be needed to detect a difference at 80% power with α = 0.05. Given the multiple comparisons in this study, the desire to explore relationships between MCC and secondary outcomes, and to guard against data loss due to subject drop-out, we planned to enroll 10 subjects.

## Randomization

Randomization was performed by study statistician, AC, using a Latin square method to determine a block of 10 sequences, which were assigned to each subject after completion of the

baseline visit. Due to the nature of the interventions, blinding was not possible. Due to the low-risk nature of this study, no interim analysis or *a priori* stop rules were planned,

## Statistical analyses

Because the purpose and design of the study was to compare the effect of different ACT modalities when added to systematic huff coughing, a non-parametric repeated measures one-way ANOVA analysis (Friedman's test) was used, as well as paired comparison to baseline huffcoughing alone for both the primary outcome and all secondary outcomes, adjusted for multiple comparisons using Dunn's test. Sample size was determined based on power calculation from other studies using these methods (S4 File). For FeNO and EBC values, pre- and postvalues from the different ACT methods were pooled and compared using Wilcoxon matchedpairs signed rank test. Spearman correlation analyses were performed to investigate any correlation between change in candidate biomarkers and MCC.

## Results

### Subjects

Ten subjects were enrolled and completed all study visits as shown in Fig 1 between January and December 2017. Subject characteristics are shown in Table 1. One subject was treated for influenza during the study period, and procedures were delayed to allow full recovery before completing the subsequent study visits. All subjects completed the study visits and tolerated study procedures well. Aside from the subject with influenza mentioned above, there were no adverse events during the study. No subject was treated with a highly effective modulator at the time of the study.

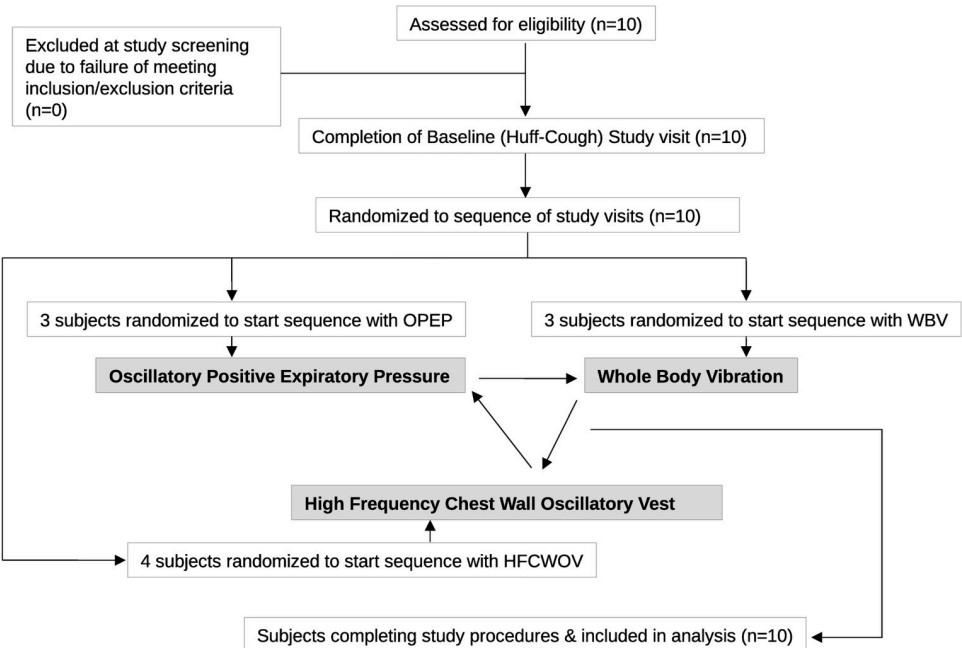

**Fig 1. Subject flow diagram.** Ten subjects were recruited and completed all study procedures. Following screening, subjects performed baseline/huff-cough measurements before being randomized to a sequence in which to complete the other studied ACT methods. All recruited subjects were included in the analysis.

**Table 1. Subject characteristics.**

| Subject | Age | Sex | FEV1 | HS | Dornase |
|---|---|---|---|---|---|
| 1 | 46 | M | 62% | Yes | Yes |
| 2 | 56 | F | 69% | Yes | No |
| 3 | 42 | F | 72% | No | No |
| 4 | 35 | F | 47% | Yes | No |
| 5 | 36 | M | 57% | Yes | No |
| 6 | 57 | F | 32% | Yes | Yes |
| 7 | 55 | F | 47% | No | Yes |
| 8 | 21 | F | 41% | No | Yes |
| 9 | 24 | F | 113% | No | Yes |
| 10 | 32 | M | 89% | Yes | Yes |

All subjects completed study procedures. Average age was 40.4 years and average percent predicted FEV1 was 62.9%.

## Mucociliary clearance and cough clearance

Average clearance of inhaled isotope through 274 minutes for each ACT method is shown in Fig 2. No apparent differences in clearance versus time curves were observed. In the primary analysis, no differences were observed in average clearance over 274 minutes across all ACT methods (Friedman's one-way nonparametric ANOVA, $p = 0.6149$), nor were any differences observed in paired comparisons between each of the three ACT methods and huff coughing alone, the baseline comparator. Similarly, no differences were observed in other MCC parameters. Because isotope deposition pattern can strongly influence subsequent clearance rates, we compared deposition skew and C/P ratio measured at each MCC study. No differences in deposition were observed between study visits that might confound interpretation of MCC rates [normalized C/P ratio ($p = 0.6685$); skew ($p = 0.0503$)]. A multiple comparisons analysis revealed a lower skew value at the OPEP visit when compared to huff-cough (1.50 vs 1.17, $p = 0.0459$). This difference was not believed to influence the primary outcome in a meaningful way, however.

## FeNO and EBC

The instrument used to measure FeNO was limited in that it was unavailable for 3 of the 40 study visits and has a lower limit of detection of 5 ppm. In 9 of the remaining 37 (24%) study visits, both the pre- and post-ACT FeNO values were below this threshold, and these results were not included in the analysis. One pre-ACT value was below sensitivity threshold with a post-ACT value above the threshold, and four post-ACT values were below sensitivity threshold with corresponding pre-ACT values above the threshold. In these cases, 4.5 ppm was used for the below-threshold value to allow for conservative comparisons. Changes in FeNO values are shown in Fig 3, and were decreased from baseline for each ACT type. Although there were insufficient samples within each ACT type to achieve significance, in a pooled comparison, there was a small decrease in FeNO following ACT (average decrease of 1.51 ppm, $p = 0.006$ using Wilcoxon signed-rank test). No correlation was identified between MCC rate and FeNO value.

Results of small molecule analysis from EBC are shown in Table 2. Pooling results from all ACT techniques, there appeared to be a small decrease in adenosine, nicotinamide, and phenylalanine following ACT. However, the veracity of these findings is unclear given the inherent difficulty in interpreting relatively weak signal strength when many variables are

**A**

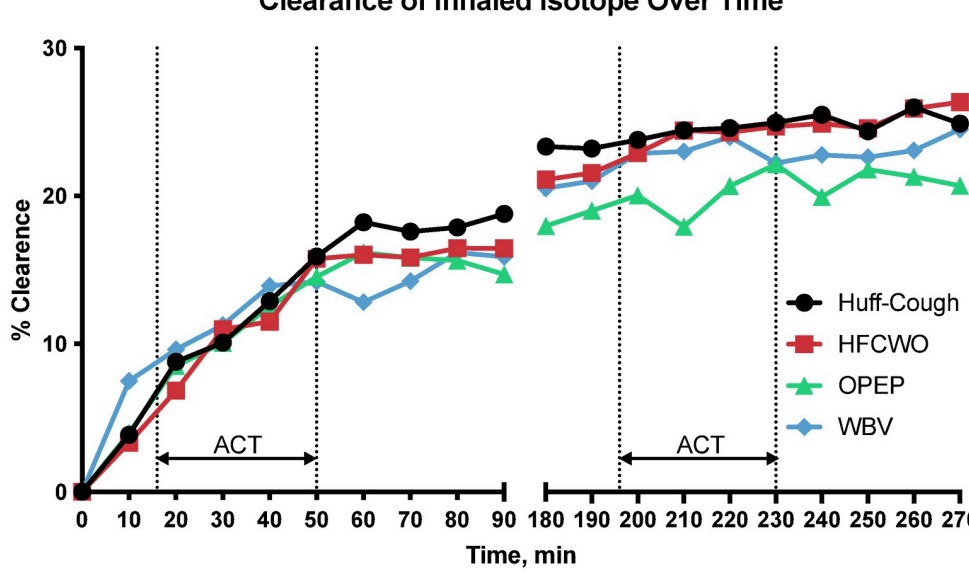

**B**

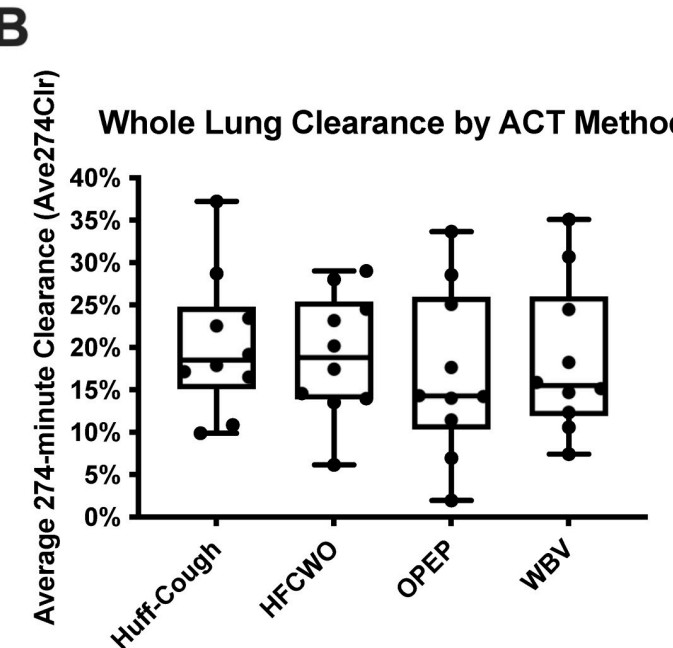

**Fig 2. Clearance of inhaled isotope as measured via gamma scintigraphy.** (A) Average clearance of inhaled isotope over time through 274 minutes. (B) Box plot of average whole lung clearance through 274 minutes. No differences were observed between ACT methods.

being studied. A larger and clearer effect was observed in lactate levels, which fell from a median value of 5.25 for the pre-ACT samples to 2.28 following ACT (p < 0.0001). Additionally, Spearman correlation analyses were performed for each candidate biomarker to investigate correlation between difference in concentration before and after ACT with MCC, as represented by Ave274Clr.

# Change in FeNO, by ACT modality

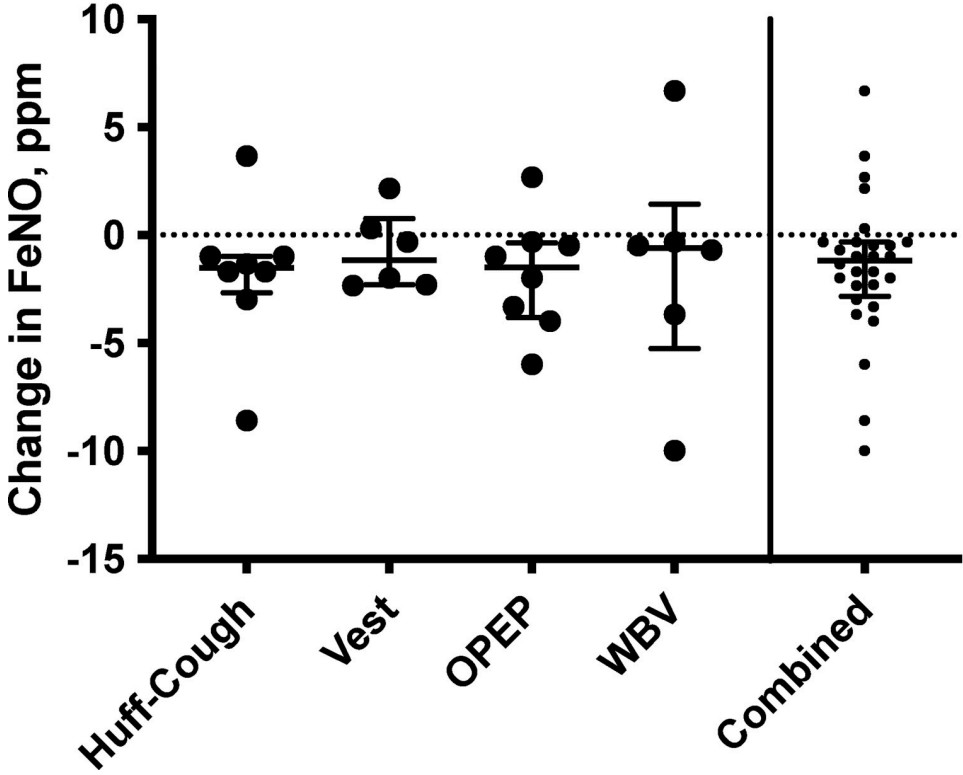

**Fig 3. Change in FeNO following ACT.** Median values and interquartile range are demarcated. Each ACT method showed an average decrease in FeNO following ACT, but this did not reach statistical significance until all values were pooled (p = 0.006, Wilcoxon signed-rank test).

Six subjects expectorated sputum during at least one study visit, with complete data for all ACT types in only four subjects. Given the paucity of data, no analysis of sputum weights or percent solids was performed, but these results are available in S6 File.

## Discussion

Although ACTs remain a central component of CF therapy, our study did not demonstrate any measurable short-term difference between ACT methods and huff-cough alone on MCC. Although surprising, these results corroborate an earlier gamma scintigraphy study [18]. It is also consistent with a general lack of difference between modalities on clinical outcomes such as spirometry and exacerbation frequency. In addition to studying the two most common ACT techniques used in the United States, we also studied whole-body vibration, a device developed for purposes of exercise enhancement rather than airway clearance. Some anecdotal reports indicated that this may be an effective method of ACT, and it allowed the opportunity to study the effects of mechanical stimuli similar to those in use in CF therapies (i.e. transduced oscillatory mechanical forces) but not specifically designed for that purpose. However, none of the ACT techniques we studied seemed to perform differently than any other or increase the effect of huff-cough alone.

The negative results from this study may be a result of a small sample size, as well as a study design that utilized huff-coughing as a control technique in place of a true baseline

**Table 2. Relative concentration of small molecules analyzed from exhaled breath condensate.**

| Small Molecule | Change in Concentration with ACT | | | Association between concentration change and MCC (Ave274Clr) | |
|---|---|---|---|---|---|
| | Pre-ACT relative concentration, median (IQR) | Post-ACT relative concentration, median (IQR) | p-value (Wilcoxon) | Spearman correlation ρ, 95% CI | p-value |
| *Purine Metabolism* | | | | | |
| Adenosine | 0.104 (0.071–0.139) | 0.0741 (0.0411–0.128) | *0.041*\* | **0.294 (-0.029–0.561)** | **0.066** |
| AMP | 0.173 (0.118–0.329) | 0.112 (0.0629–0.270) | 0.146 | 0.119 (-0.210–0.422) | 0.466 |
| Hypoxanthine | 0.599 (0.334–1.11) | 0.434 (0.204–0.876) | 0.070 | -0.005 (-0.325–0.315) | 0.973 |
| Nicotinamide | 0.341 (0.258–0.532) | 0.290 (0.200–0.437) | *0.030*\* | **-0.033 (-0.350–0.290)** | **0.838** |
| Uric Acid | 0.00228 (0.00108–0.00402) | 0.00243 (0.00115–0.00523) | 0.7446 | 0.038 (-0.286–0.354) | 0.816 |
| *Nucleosides* | | | | | |
| Cytidine | 0.0987 (0.0795–0.117) | 0.0710 (0.0369–0.128) | 0.183 | 0.201 (-0.127–0.490) | 0.214 |
| Inosine | 0.142 (0.0854–0.244) | 0.128 (0.0471–0.263) | 0.765 | 0.052 (-0.273–0.366) | 0.749 |
| *Amino Acids / Peptides* | | | | | |
| Isoleucine | 2.41 (1.40–6.37) | 1.53 (0.927–4.69) | 0.128 | -0.037 (-0.353–0.286) | 0.820 |
| Leucine-proline dipeptide | 2.50 (1.69–3.18) | 2.07 (0.934–3.00) | 0.206 | 0.046 (-0.279–0.360) | 0.780 |
| Phenylalanine | 5.50 (4.33–8.96) | 4.73 (2.60–8.01) | *0.041*\* | **0.005 (-0.325–0.316)** | **0.976** |
| Uridine | 0.469 (0.299–0.772) | 0.349 (0.223–0.827) | 0.647 | 0.105 (-0.223–0.411) | 0.521 |
| *Polyamine Synthesis Pathway* | | | | | |
| Spermine | 0.0855 (0.0741–0.114) | 0.0782 (0.0664–0.104) | 0.438 | 0.173 (-0.155–0.468) | 0.285 |
| *Cellular Energy Metabolites* | | | | | |
| Lactate | 5.25 (0.625–7.51) | 2.28 (1.09–5.67) | **<0.0001**\*\*\* | **0.313 (-0.008–0.575)** | **0.050** |
| *Mucin Components* | | | | | |
| Sialic Acid | 0.00159 (0.000698–0.00583) | 0.00176 (0.00133–0.00272) | 0.196 | 0.017 (-0.305–0.335) | 0.918 |

Small molecule concentration changes with ACT as well as correlation to MCC are reported. Median values and inter-quartile ranges for the relative concentrations of each candidate biomarker pre- and post-ACT are given. In order to correct for variability of dilution of samples, concentrations were normalized to urea concentration and are reported as a unitless ratio. Pre- and post-ACT samples were compared using a Wilcoxon signed rank test. Association of concentration change to Ave274Clr was investigated using Spearman correlation, ρ. These values are reported with 95% confidence interval as well as two-tailed p-value.

measurement. Since other ACT modalities are intended to potentiate the effect of cough-based clearance, our study design included huff-coughs to be performed with each of the other modalities. It is possible that adjunctive ACT devices do not have an effect on mucus clearance beyond huff-cough alone, however, the collective experience in CF treatment strongly advocates for the benefits of the use of adjunctive airway clearance modalities, so this conclusion seems unlikely. Alternatively, the timing between ACT treatment and MCC measurement may also have been inappropriate to detect important biological changes.

Furthermore, it may be that ACT does not contribute to mucus clearance in a manner which can be appreciated using the MCC measurement technique utilized in this study. For example, it may be that the greatest effect of ACT is to open airways obstructed by mucus plugs, an effect not detectable with this assay. Since subjects inhaled the radiotracer prior to the initiation of ACT, no isotope deposition would have occurred in airways lacking ventilation (i.e. completed obstructed). As a result, opening of these airways would not be detected.

Our secondary outcomes did reveal some unexpected findings. We included FeNO as an outcome measure because it is known to be lower in individuals with CF compared to healthy individuals, and has been associated with disease activity, and was improved in individuals treated with highly effective CFTR modulators [20, 26]. NO is known to act through guanylate

cyclase to increase ciliary beat frequency, and thus may promote or reflect increased MCC [27, 28]. Indeed, this relationship was demonstrated in sheep, and whole-body vibration was implicated to accelerate MCC via increased NO release [29]. We anticipated that with stimulated MCC via ACT, FeNO would increase, but instead found a small, but opposite, effect. We speculate that this may be due to the opening of plugged airways with greater disease activity and lower levels of NO.

In vitro studies of the role of small molecules in airway biology, particularly with regard to the regulation of hydration of the airway surface liquid and the function of the mucociliary apparatus led us to hypothesize that mechanical forces applied to the airways during ACT would lead to an increase in ATP release from epithelia and appearance of purine metabolites (AMP and adenosine, in particular) in EBC following ATP metabolism [30–32]. However, we observed no such increases, and even a possible decrease in two purines. These results may suggest that ACT does not stimulate MCC through endogenous regulatory pathways, but rather may confer benefit through other mechanisms, such as mechanical disruption of mucus. The most striking result was with the marked decline of lactate in EBC, which decreased with ACT on average about 25%. While the cause of this change isn't readily apparent from our study, this could represent improved aeration and oxygenation of plugged small airways with more aerobic metabolic activity in those diseased airways following ACT. A previous study of BALF in young children with CF showed a close positive correlation between mucin concentration and lactate in BALF, which was interpreted as mucin plugs causing localized hypoxia [33]. In addition to a rather small sample size, another limitation of the EBC analysis in our study was that FeNO measurements were taken first following ACT, and with three replicate FeNO values obtained with time required for the device to process each sample, the EBC collection did not begin until at least five minutes following completion of ACT, which may be adequate time for normalization for some of these small molecules.

We do not believe our results constitute sufficient grounds to recommend a change in clinical practice as directed by consensus guidelines in the US, UK, Europe, and Australia. However, these findings do suggest that ACT does not confer clinical benefit in the same manner as therapies which have been shown to increase MCC, such as hypertonic saline and ivacaftor for gating mutations. Future studies should include a true baseline of MCC as well as alterations in study design which may explore whether ACT has an effect on patency of airways, particularly in the era of highly effective modulators. This could be accomplished with post-ACT aerosol deposition or ventilation imaging by gamma scintigraphy or MRI. This work would be important to confirm the presumed clinical benefits of ACT and continue to justify the significant treatment burden they place on individuals with CF and their families.

## Supporting information

**S1 Checklist. CONSORT checklist.** Completed CONSORT checklist.
(DOC)

**S1 File. Mucociliary clearance data obtained from gamma scintigraphy.** Excel spreadsheeting containing MCC data used for analysis including retention and clearance in all ROIs, with deposition data (skew, C/P).
(XLSX)

**S2 File. FeNO data used in analysis.** Excel spreadsheeting containing pre- and post-ACT FeNO data for each subject and each ACT modality.
(XLSX)

**S3 File. Small molecule concentrations in exhaled breath condensate.** Excel spreadsheet containing pre- and post-ACT small molecule concentrations for each subject and each ACT modality. The raw concentrations are given, as well as those normalized to urea concentration, as was used in analysis.
(XLSX)

**S4 File. Study protocol.** Study protocol prepared *a priori*, as reviewed by the institutional board.
(PDF)

**S5 File. Further methods describing timing during study visits as well as additional details of the standardized methods used for MCC measurement.**
(DOCX)

**S6 File. Data on sputum weights (in g) and % solids collected from subjects during study visits.** These were insufficient to be used in analysis.
(XLSX)

## Acknowledgments

The authors would like to thank Alexandra Nesbitt and Amy Brightwood Gordon for their assistance in conducting this study.

## Author Contributions

**Conceptualization:** Aaron Trimble, William Bennett, Scott Donaldson.

**Data curation:** Aaron Trimble, Kirby Zeman, Jihong Wu, William Bennett, Scott Donaldson.

**Formal analysis:** Aaron Trimble, Kirby Zeman, Jihong Wu, Agathe Ceppe, William Bennett, Scott Donaldson.

**Funding acquisition:** Aaron Trimble, Scott Donaldson.

**Investigation:** Aaron Trimble, Kirby Zeman, Jihong Wu, William Bennett, Scott Donaldson.

**Methodology:** Aaron Trimble, Kirby Zeman, Agathe Ceppe, William Bennett, Scott Donaldson.

**Project administration:** Aaron Trimble, Kirby Zeman, Jihong Wu, William Bennett, Scott Donaldson.

**Resources:** Aaron Trimble, Kirby Zeman, Jihong Wu, William Bennett, Scott Donaldson.

**Software:** Aaron Trimble, Kirby Zeman, Jihong Wu, Agathe Ceppe, William Bennett, Scott Donaldson.

**Supervision:** Aaron Trimble, William Bennett, Scott Donaldson.

**Validation:** Aaron Trimble, Kirby Zeman, Jihong Wu, Agathe Ceppe, William Bennett, Scott Donaldson.

**Visualization:** Aaron Trimble, Kirby Zeman, Jihong Wu, Agathe Ceppe, William Bennett, Scott Donaldson.

**Writing – original draft:** Aaron Trimble, Scott Donaldson.

**Writing – review & editing:** Aaron Trimble, Kirby Zeman, Jihong Wu, Agathe Ceppe, William Bennett, Scott Donaldson.

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
