## [Decision Letter · Decision Letter 0]

8 Jul 2021

PONE-D-21-05105

Effect of Airway Clearance Therapies on Mucociliary Clearance in Adults with Cystic Fibrosis: A randomized controlled trial

PLOS ONE

Dear Dr. Trimble,

Thank you for submitting your manuscript to PLOS ONE. After careful consideration, we feel that it has merit but does not fully meet PLOS ONE’s publication criteria as it currently stands. Therefore, we invite you to submit a revised version of the manuscript that addresses the points raised during the review process.

A major concern raised by the reviewers is the sample size. They felt that the sample size of 10 subjects was small, and that you did not provide a compelling justification for your determination of this as an ideal sample size. Please note that further consideration of your manuscript will depend on your ability to provide a clear and compelling justification for your methodology, including the sample size. They also raised concerns with your reasoning and justification for the statistical analysis. For example, how do you justify using linear regression to establish correlation, with a sample size of n = 10?They also felt that you should use non-parametric analysis for all variables. Their comments can be viewed, in full, below. 

We look forward to receiving your revised manuscript.

Kind regards,

Natasha McDonald, PhD

Associate Editor

PLOS ONE

Journal Requirements:

2. Thank you for submitting your clinical trial to PLOS ONE and for providing the name of the registry and the registration number. The information in the registry entry suggests that your trial was registered after patient recruitment began. PLOS ONE strongly encourages authors to register all trials before recruiting the first participant in a study.

1) your reasons for your delay in registering this study (after enrolment of participants started);

2) confirmation that all related trials are registered by stating: “The authors confirm that all ongoing and related trials for this drug/intervention are registered”.

3. Please include your statement regarding patient consent in the manuscript Methods.

4. Thank you for stating the following in the Financial Disclosure section:

"This work was supported by a grant to AT and SD from the Cystic Fibrosis Foundation (http://www.cff.org REACT16A0). The funders of this study were not involved in the study design, execution of the procedures, data analysis, preparation of the manuscript, or decision to publish. Investigators received no financial support from commercial entities for the completion of this study. A PowerPlate® device was provided by the manufacturer, Performance Health Systems LLC, Northbrook, IL, USA."

We note that you received funding from a commercial source: Performance Health Systems LLC.

Reviewers' comments:

Reviewer's Responses to Questions

**Comments to the Author**

1. Is the manuscript technically sound, and do the data support the conclusions?

Reviewer #1: Yes

Reviewer #2: Yes

Reviewer #3: Yes

2. Has the statistical analysis been performed appropriately and rigorously? 

Reviewer #1: No

Reviewer #2: No

Reviewer #3: Yes

3. Have the authors made all data underlying the findings in their manuscript fully available?

Reviewer #1: Yes

Reviewer #2: No

Reviewer #3: Yes

4. Is the manuscript presented in an intelligible fashion and written in standard English?

Reviewer #1: Yes

Reviewer #2: Yes

Reviewer #3: Yes

5. Review Comments to the Author

Reviewer #1: The objective of this study is to investigate the effectiveness of airway clearance therapy on mucociliary clearance (MCC) adults with cystic fibrosis (CF). The authors considered an open-label, multiple cross-over study study. The study was approved by the Institutional Review Board at Univ. of North Carolina, and carries a legitimate NCT number. While the study objectives sound interesting, a number of shortcomings were observed, in regards to abiding by the CONSORT guidelines for conducting and reporting results of high-quality randomized clinical trials (RCTs), as well as some statistical and reporting issues:

1. Abstract:

It's better to present the Abstract in accordance to the 4 sections: Background, Methods, Results and Conclusions. Also, in the Results section, statistical significance/non-significance of findings should be expressed with p-values, and estimated precision, such as confidence intervals, or CIs (say, 95\\%). Check CONSORT checklist for Abstracts reporting of RCTs.

2. Methods:

Methods reporting appeared very messy. An orderly manner is suggested, following CONSORT guidelines, without repeating information, such as Trial Design, Participant Eligibility Criteria and settings, Interventions, Outcomes, sample size/power considerations, Interim analysis and stopping rules, Randomization (details on random number generation, allocation concealment, implementation), Blinding issues, etc. The authors are advised to create separate subsections for each of the possible topics (whichever necessary), and that way produce a very clear writeup. I see the Authors indeed made an attempt; however, they are advised to write it carefully, following nice examples in the manuscript below:

https://www.sciencedirect.com/science/article/pii/S0889540619300010

(a) For instance, the randomization technique employed (Latin square method) looks almost hidden; it should be pointed clearly. Furthermore, randomization and allocation concealment should be made very clear; the trial staff recruiting patients should not have the randomization list. Randomization should be prepared by the trial statistician, and he/she would not participate in the recruiting.

(b) I am surprised to see no statement on sample size/power in a manuscript proposing a cross-over trial. A vague statement was provided as "Sample size was determined based on..", with no specific determination of sample size wrt. some desired effect size. This is really a key here; else one has no idea why the study recruited and analyzed only 10 subjects.

(c) Friedman's one-way nonparametric ANOVA test was used, which looks OK. However, to assess correlation between changes in candidate biomarkers, the authors stated a linear regression was used. From my understanding of cross-over trials, a repeated measures analysis, via linear mixed-models is more appropriate. In that context, it is doubtful how much credible the study findings will be with only 10 subjects!

3. Results & Conclusions:

The authors ignored a thoughtful discussion of the presence/absence of a desirable 'washout' period, which is the cornerstone of a cross-over trial. The sample size enrolled is somewhat small, reflecting a specific population. I can only consider this study, at best, as a very pilot study. However, with no clear description of how they came up with 10 subjects, I am hesitant to proceed further.

For understanding and designing a nice multi-crossover study, the authors are asked to follow this article below (which also discusses the utility of using random-effects models, or the linear/nonlinear mixed-effects models):

https://www.ncbi.nlm.nih.gov/pmc/articles/PMC6481185/

Reviewer #2: Thank you for the opportunity for reviewing this exciting manuscript including non-conventional outcome measures to assess the effects of different ACTs at short-term in people with cystic fibrosis. I really enjoyed reading it and learned as well. However, I would like to discuss some points with the authors.

a. The timeline of the study is a bit confusing and not easy to follow-up at first glance. Could you please try to explain it in a simpler way? You chose a complex procedure and it may be difficult for future readers to follow, especially if they are not familiar with this type of outcome measures

b. Why did you assess the effects of two ACT sessions, one followed by the other? Although two sessions per day are recommended in clinical practice, they are not back-to-back.

c. What is the rationale for assessing the whole effects of ACTs only up to 40 minutes after the end of the second session? The possibility of seeing differences over a longer period has been lost.

d. I was surprised that the autogenic drainage technique was not included in the study, as it is one of the most popular techniques in this population. Why do not include techniques that increase expiratory airflow by reducing the cross-sectional ratio of the airways when they seem to be best suited according to the mucus clearance physiology?

e. I do not agree to compare the body vibration device with exercise. The main mechanism of exercise that improves airway clearance is increased airflow. However, a body vibration device could be more similar to HFCWO device.

f. Why were the pulmonary biomarkers (FeNO and EBC) only obtained after the first ACTs session? Why not also after the second ACTs session like the MCC?

g. According to table 1, six participants use daily hypertonic saline as part of their routine treatment. Did the participants take the HS the days of the study? If so, when did they take it? How do you control this factor? Could it have influenced your results?

h. Were the participants adherent/trained to any ACTs? What were the ACTs they routinely used?

i. Could you please give us more information on the daily sputum expectoration of your participants? Alternatively, any information to understand better the level of impairment of the airway clearance in your sample?

j. I believe that huff-cough manoeuvres is a good ´comparator´ in this study. However, how many standardised cough manoeuvres performed the participants in each study visit? How many spontaneous cough manoeuvres were recorded in each study visit? Were this number similar between the different treatment arms? No data were reported in results.

k. Although no difference was observed between the treatment arms; OPEP showed a tendency to clear less Tc99m at the end of observed period (270 min). OPEP is the only ACTs analysed in this study that requires the active collaboration of the patient. It is described in the manuscript that the three ACTs were instructed to participants at the screening visit, however, were the participants able to test the OPEP device?; if participants were not sufficiently familiar with these devices, could lack of experience have affected your results?

l. Do you really think that FeNO is an appropriate outcome measure to assess the effects of ACTs in your sample considering that almost 25% of your measures did not achieve the minimum threshold? Please, could you explain this point in the discussion section? In addition, could you explain why there is an opposite direction of change in your study in this outcome measure?

m. I checked the information provided in the supplementary material and was surprised that you did not include any information on sputum quantity or % of solids in this manuscript. However, both outcomes seem to be assessed according to your initial protocol (supplementary material). Why did you not include this information in the manuscript?

n. Your data analysis is a bite confusing to me.

a. First, you suggest using non-parametrical approach in methods. I agree with this approach because of your sample size and the non-normal distribution of your outcome measures. However, it seems that you use parametric tests in the results section (e.g., page 11, line 225 it is stated that you use ANOVA; it is explained in figure 4 and in page 12, line 248 that you use t-student for paired sample; you describe your results in the figures using mean instead of median). For that reason, I recommend performing again the statistical analysis using a non-parametrical approach for all your variables and describe the findings using median and P25-P75, as you correctly did for EBC findings in table 2.

b. Why not include the effect size calculation in the results obtained from EBC? It would be useful to understand better the magnitude of the change.

c. It seems to me (based on your data and graphs) that you calculate correlations of two variables rather than linear regressions models. If this is correct, please change the information provided in the statistical analysis (page 10, line 198) and page 13, line 261) and give us information about the value of the correlation (r value) with their confidence intervals at 95%. If you finally calculate linear regressions model (I do not think it is appropriate for your sample size and the distribution of your outcome measures), you need to give us information about the odds ratio and the confidence intervals of your outcome measures.

o. Table 2 shows several molecules that have not been mentioned in the introduction or in the methods section. Could you please introduce a short paragraph to explain the role of these molecules in the airway clearance and guide to the readers?

In addition, you will find my minor comments / suggestions in the following table:

Page number Sentence / Section Issues which could be considered

Page 3, line 42 Introduction Suggest removing the word ´mechanical´ because the information is described in a general way.

Page 3, line 52 Introduction Authors referred to a SR published in 2017; however, it is not the latest version. Could the authors please check and change the information provided according to the updated version of this SR? (Cochrane Database Syst Rev. 2020 Apr 30;4(4):CD006842.

Page 3, line 42 Introduction Suggest removing the word ´mechanical´ because postural drainage and exercise are not theoretically considered mechanical ACTs.

Page 4, line 64 Introduction From my point of view, the introduction section is not the best place to describe the methodology of your study. Thus, I suggest removing the information included from line 64 to the end of the paragraph. Instead of this information, the authors may i) describe the different mechanism of action of the three techniques explored in the study; ii) or even explain why the biomarkers assessed are appropriate for evaluating the effects of ACTs because they are poorly explored in this type of studies.

Page 4, line 64 Introduction Suggest removing the word ´widely-used´ because HFCWO is not as commonly used in Europe or Asia as in the USA in this population.

Page 4, line 75 Hypothesis Please, clarify that MCC/CC was your primary endpoint and the rest of outcome measures were considered secondary endpoints.

Page 5, line 86 Methods How long was the washout period between the treatment arms? Was the washout period standardised?

Page 7, line 130 Methods Why did the authors increase the frequency each cycle? Why did the authors decide to set different pressure in each cycle? What is the reason for using this protocol?

Page 7, line 138 Methods Please, clarify that the number of coughs were registered in all clinic visits and not only in the last one.

Page 10, table 1 Results Did the participants have a chronic airway infection? Please, include additional information on possible pathogens in table 1

Page 12, fig 3 (B); fig 4 Results Please, change ´Vest´ to HFCWO

Page 14, line 276 Discussion Suggest including at short-term at the end of the sentence because you did not explore longer-term effects (24h for example).

Page 14, line 292 Discussion Suggest being caution with this sentence. In your study, you did not find any difference between ACTs plus huff-cough and huff-cough alone at very short-term and did not evaluate the most popular ACTs in this population. This does not mean that the techniques may not provide any clinical benefit in this population

Page 15, line 300 Discussion The limitation described for MCC is similar to lung clearance index to assess the effects of ACTs. Could you describe/name it?

Page 16, line 318 Discussion Suggest including your small sample size as a limitation to find a more reliable results in purines and the other small molecules.

Reviewer #3: Comments to Author:

Trimble and colleagues present a well-written paper addressing differences in airway clearance therapies on mucociliary clearance in people with CF. Although a similar study has been previously conducted with a similar sample size, the authors were able to rigorously study their research question in a well-designed study and include more current therapies and devices. Additionally, the authors were able to incorporate a novel secondary aim of evaluating small molecules as potential biomarkers of ACT clearance. Although the results of the study were negative, the question remains important and a topic for future research given widespread use of CFTR modulators and the need for individualized ACT therapies based on patient preference and cost effectiveness.

Title: No changes to suggest

Abstract: Incorporates main points concisely, no major changes to suggest

Intro:

The authors nicely provide relevant background while stating the primary objective of assessing for efficacy of ACT methods on mucociliary clearance.

P3, line 57) If word limit allows, would consider adding a sentence on how gamma scintigraphy works given the wider range of the PLOS audience (ie: technique that combines administration of radioistopes in conjunction with imaging modalities, etc).

Methods:

P5, line 85-86) Would ask authors to consider further describing timing of study visits, mentions 4 study visits but the timing of visits is not understood

Results:

P10, line 205) Would authors be able to elaborate on the number of participants/readings the FENO device malfunctioned? **It is found further down in results which would be fine to leave it placed there

Discussion:

Authors could consider commenting on small sample size of study, as it was similar to the prior study referenced in their article (ref 18). Otherwise, authors appropriately discuss limitations, notably obtaining a true baseline as the huffing group may have impacted their results. Given this study was performed prior to widespread use of CFTR modulators, it would be interesting to also include exercise as a group in future studies given that many people with CF have self-discontinued use of some ACT devices given overall improvement in clinical health.

6. PLOS authors have the option to publish the peer review history of their article (what does this mean?). If published, this will include your full peer review and any attached files.

Reviewer #1: No

Reviewer #2: **Yes: **Beatriz Herrero-Cortina

Reviewer #3: No

---

## [Author Response · Author response to Decision Letter 0]

21 Aug 2021

Our point-by-point responses to the reviewers is provided in the attached rebuttal letter. Our responses to the specific comments from Journal Editor Dr. Natasha McDonald in the decision letter is below:

1) We have updated the manuscript to meet the style requirements as requested.

2) We have updated the manuscript's method section to note the reason for inclusion in registry following subject enrollment (due to institutional delays with PRS system)

3) We have updated the methods section to note patient consent.

4) With regard to potential competing interests, we would like to clarify that the device we received from the commercial entity was provided to us temporarily, and was returned following completion of the study, and we therefore do not see the loan of this device as a competing interest, and we stand by our original statement with regard to competing interests. However, we would like to append the following to our original financial disclosure statement:

“This device was returned to the manufacturer following completion of the study.”

However, if the editorial board does not share this view with regard to our competing interests we provide competing interests statement to as follows:

“The authors received a PowerPlate® device from the manufacturer (Performance Health Systems LLC, Northbrook, IL, USA). This product was provided for the purposes of conducting this study without stipulation or conditions, and returned following completion of the study. No other goods, services, or any other form of compensation were provided to the authors by this entity. This does not alter our adherence to PLOS ONE policies on sharing data and materials. ”

---

## [Decision Letter · Decision Letter 1]

5 May 2022

Effect of airway clearance therapies on mucociliary clearance in adults with cystic fibrosis: a randomized controlled trial

PONE-D-21-05105R1

Dear Dr. Trimble,

We’re pleased to inform you that your manuscript has been judged scientifically suitable for publication and will be formally accepted for publication once it meets all outstanding technical requirements.

Kind regards,

D William Cameron, MD

Academic Editor

PLOS ONE

Additional Editor Comments (optional):

Reviewers' comments:

Reviewer's Responses to Questions

**Comments to the Author**

1. If the authors have adequately addressed your comments raised in a previous round of review and you feel that this manuscript is now acceptable for publication, you may indicate that here to bypass the “Comments to the Author” section, enter your conflict of interest statement in the “Confidential to Editor” section, and submit your "Accept" recommendation.

Reviewer #1: All comments have been addressed

Reviewer #2: All comments have been addressed

2. Is the manuscript technically sound, and do the data support the conclusions?

Reviewer #1: (No Response)

Reviewer #2: Yes

3. Has the statistical analysis been performed appropriately and rigorously? 

Reviewer #1: (No Response)

Reviewer #2: Yes

4. Have the authors made all data underlying the findings in their manuscript fully available?

Reviewer #1: (No Response)

Reviewer #2: Yes

5. Is the manuscript presented in an intelligible fashion and written in standard English?

Reviewer #1: (No Response)

Reviewer #2: Yes

6. Review Comments to the Author

Reviewer #1: (No Response)

Reviewer #2: Many thanks for addressing all my comment s/ suggestions. Please continue research in this field and including other chronic respiratory diseases.

7. PLOS authors have the option to publish the peer review history of their article (what does this mean?). If published, this will include your full peer review and any attached files.

Reviewer #1: No

Reviewer #2: **Yes: **Beatriz Herrero Cortina

---

## [Editor Report · Acceptance letter]

12 May 2022

PONE-D-21-05105R1 

Effect of airway clearance therapies on mucociliary clearance in adults with cystic fibrosis: a randomized controlled trial 

Dear Dr. Trimble:

I'm pleased to inform you that your manuscript has been deemed suitable for publication in PLOS ONE. Congratulations! Your manuscript is now with our production department. 

Kind regards, 

on behalf of

Professor D William Cameron 

Academic Editor

PLOS ONE